# Drug Resistance in Glioma Cells Induced by a Mesenchymal–Amoeboid Migratory Switch

**DOI:** 10.3390/biomedicines10010009

**Published:** 2021-12-22

**Authors:** Sophie E. Ketchen, Filomena O. Gamboa-Esteves, Sean E. Lawler, Michal O. Nowicki, Arndt Rohwedder, Sabine Knipp, Sally Prior, Susan C. Short, John E. Ladbury, Anke Brüning-Richardson

**Affiliations:** 1Light Laboratories, School of Molecular and Cellular Biology, University of Leeds, Leeds LS2 9JT, UK; s.e.ketchen@leeds.ac.uk (S.E.K.); A.Rohwedder@leeds.ac.uk (A.R.); J.E.Ladbury@leeds.ac.uk (J.E.L.); 2Leeds Institute of Medical Research at St. James’s, University of Leeds, Leeds LS9 7TF, UK; F.E.O.Esteves@leeds.ac.uk (F.O.G.-E.); s.c.short@leeds.ac.uk (S.C.S.); 3Brown University Cancer Center, Pathology & Laboratory Medicine, Brown University, Providence, RI 02903, USA; sean_lawler@brown.edu; 4Harvey Cushing Neuro-Oncology Research Laboratories, Department of Neurosurgery, Brigham and Women’s Hospital, Harvard Medical School, Boston, MA 02115, USA; mnowicki@bwh.harvard.edu; 5School of Applied Sciences, University of Huddersfield, Huddersfield HD1 3DH, UK; S.Knipp@hud.ac.uk (S.K.); u1973968@unimail.hud.ac.uk (S.P.)

**Keywords:** migration, invasion, glioblastoma, CCN1, mesenchymal–amoeboid transition, biomarker

## Abstract

Cancer cell invasion is a precondition for tumour metastasis and represents one of the most devastating characteristics of cancer. The development of drugs targeting cell migration, known as migrastatics, may improve the treatment of highly invasive tumours such as glioblastoma (GBM). In this study, investigations into the role of the cell adhesion protein Cellular communication network factor 1 (CCN1, also known as CYR61) in GBM cell migration uncovered a drug resistance mechanism adopted by cells when treated with the small molecule inhibitor CCG-1423. This inhibitor binds to importin α/β inhibiting the nuclear translocation of the transcriptional co-activator MKL1, thus preventing downstream effects including migration. Despite this reported role as an inhibitor of cell migration, we found that CCG-1423 treatment did not inhibit GBM cell migration. However, we could observe cells now migrating by mesenchymal–amoeboid transition (MAT). Furthermore, we present evidence that CCN1 plays a critical role in the progression of GBM with increased expression in higher-grade tumours and matched blood samples. These findings support a potential role for CCN1 as a biomarker for the monitoring and potentially early prediction of GBM recurrence, therefore as such could help to improve treatment of and increase survival rates of this devastating disease.

## 1. Introduction

Localisation and the invasive behaviour of the most aggressive form of gliomas, glioblastoma (GBM), contributes to the challenge in treating brain tumour patients, in fact, there is currently no curative approach available. Recent trends in GBM research aim to provide a better understanding of the molecular pathology and genomic landscape of GBM, which will enable tailored therapeutic approaches. Though much progress has been made in cancer treatments in general, substantial challenges remain for GBM treatment including tumour location, relapse, tumour heterogeneity and drug resistance. The current standard of care consists of surgical resection followed by radiotherapy and adjuvant chemotherapy with temozolomide [1]. Unfortunately, despite these therapeutical approaches, nearly all patients will develop a recurrent tumour and treatment options become limited to palliative care [2]. It is known, that the cell adhesion protein CCN1 (also known as cysteine-rich angiogenic inducer 61 or CYR61) stimulates mesenchymal migration in fibroblasts, endothelial cells, smooth muscle cells and some types of cancer cells [3]. Studies have also shown that CCN1 is associated with cancer progression and invasion [4,5]. The tight association between CCN1, the extracellular matrix (ECM), and the cell surface stimulates adhesive signalling and supports cell adhesion [3]. This results in the formation of focal adhesions and activation of focal adhesion kinase (FAK); paxillin and Rac are induced leading to reorganisation of the actin cytoskeleton and lamellipodia and filopodia formation [6,7,8]. 

In tumour progression, CCN1 has been shown to promote invasion, angiogenesis, cell proliferation, cell survival and metastasis [3]. Silencing CCN1 expression in vitro leads to loss of invasion [9]. In addition, in immunodeficient mice implanted with human gastric, naturally CCN1 deficient, adenocarcinoma cells induced expression of the CCN1 cDNA is an effective promoter of angiogenesis, concomitant with tumour vascularisation and growth [10,11]. CCN1 is highly expressed in primary gliomas, as well as in high-grade glioma cell lines [12]. High expression levels were also observed in the astrocytoma cell lines U87, U373 and T98G and only low expression levels in the less invasive U343 cell line. Forced expression of CCN1 in U343 cells increased proliferation, angiogenesis and tumorigenicity in vivo. U343 cells overexpressing CCN1 migrated more readily and produced larger, more vascularised tumours in nude mice [12].

Here, we used the small molecule inhibitor CCG-1423 to investigate the effects of CCN1 on cell migration in GBM cell lines. CCG-1423 blocks the production of CCN1. It binds to MKL1, preventing its nuclear import and therefore activation of MKL1/SRF-dependent transcription and CCN1 production [13]. Contrary to other studies, our in vitro results did not show CCG-1423 to have an anti-migratory effect as previously suggested [13,14]. In a recent publication, using iSIM microscopy to analyse cell morphology during migration we could show that GBM cells can respond to CCG-1423 treatment by switching their migratory mode [15]. Therefore, since our previous study indicated that CCN1 is required for mesenchymal migration, we hypothesise that glioma cells can overcome the anti-migratory drug treatment by inducing a mesenchymal–amoeboid transition (MAT) for continued migration and invasion. We also present clear evidence that CCN1 has a pivotal role in the progression of GBM, as we present evidence that its increased expression in higher grade tumours is correlated with higher concentrations in patient blood. This could provide a valuable molecular marker for the characterisation, progression, and possibly early detection of GBM. Furthermore, we present evidence that the cellular phenomenon of MAT is in fact being utilised by cancer cells as a drug resistance mechanism. This enables the cells to ‘side-step’ the action of drugs that only target one type of migration, allowing continued spread and invasion.

## 2. Materials and Methods

### 2.1. Cell Culture

The glioma cell lines U87 (U-87MG, RRID:CVCL_0022) and U251 (U-251MG, RRID:CVCL_0021) were obtained from ATCC and ECACC. U87-NT cell lines were created to stably express a GFP control construct, a kind gift from Dr Chiara Galloni (University of Leeds). The primary human oligodendroglioma cell line HOG (RRID:CVCL_D276) (gifted by Diane Jaworski, University of Vermont) and adult glioma cell line G44 [16], the murine glioma cell line GL261 (RRID:CVCL_Y003) and the rat glioma cell line CNS1 (RRID:CVCL_5276) were all maintained and grown in the Neurosurgery department at Harvard Medical School, Brigham and Women’s Hospital. U251, U87, GL261 and CNS1 cell lines were grown in Dulbecco’s Modified Eagle’s Medium (Sigma, Poole, Dorset, UK) with 10% heat-inactivated foetal calf serum (Sigma, Poole, Dorset, UK) and 0.5% penicillin-streptomycin (Sigma, Poole, Dorset, UK) (complete medium) and cultured in a Sanyo CO_2_ incubator at 37 °C in a humidified atmosphere of 5% CO_2_ (in air).

HOG and G44 were grown as neurospheres suspended in neurobasal medium (ThermoFisher, Altrincham, UK) containing B27 supplement, 0.5% penicillin-streptomycin (ThermoFisher, Altrincham, UK), glutamine (ThermoFisher, Altrincham, UK), EGF and FGF (ThermoFisher, Altrincham, UK). 

All human cell lines were recently authenticated using short tandem repeat profiling by the University of Leeds and Harvard Medical School (July 2021), and all experiments were performed with mycoplasma-free cells.

### 2.2. Tissue Microarray (TMA) Staining

A commercially available brain cancer tissue microarray (TMA) (Biomax BS17015a, Derwood, MD 20855, USA) including 63 cases/cores was stained for CCN1. The 63 cores included 38 astrocytoma grades 1–3, 14 GBM, 6 oligodendroglioma, 1 ependymoma, 1 medulloblastoma, 3 cases of cancer adjacent brain tissue (negative control) and 1 pheochromocytoma (adrenal gland tumour, positive control). Because of the small sample size of the ependymoma and medulloblastoma, these were not included in the data analysis. The TMA slide was dewaxed and underwent antigen retrieval and staining with the primary antibody anti-CYR61/CCN1 Rb (Abcam, Cambridge, UK) at 1:250 following the protocol described by Cheng et al. [17]. The slide was analysed by calculating and combining two sub-scores for the CCN1 staining: staining intensity and percentage of core stained. Staining intensity was scored out of 3 with 0 being no staining and 3 being strong staining. Percentage of staining was scored out of 4 with 0 being 0% stained, 1 being 1–25% stained, 2 being 26–50% stained, 3 being 51–75% stained and 4 being 75%+ stained. Thus, the final maximum combined score is 7.

### 2.3. Immunohistochemistry (IHC) of Patient Samples

Patient GBM samples were collected, processed and stored by the Leeds Multidisciplinary Routine Tissue Banking (RTB) service, obtained from GBM patients undergoing surgery at the Leeds General Infirmary (ethical approval no. RTB 15/YH/0080). Tumour tissues were embedded in paraffin wax, sectioned on a microtome (Leica Biosystems, Newcastle Upon Tyne, UK) at 8 µm and dried on glass microslides for 24 h. The slides were deparaffinized and rehydrated following standard procedures, then dewaxed by 2× 1-min washes in xylene (Sigma, Poole, Dorset, UK), followed by 2× 1 min washes in ethanol (Sigma, Poole, Dorset, UK) and 3× 1 min washes in water. Antigen retrieval was performed using Tris EDTA, pH 9 (Abcam, Cambridge, UK) in a pressure cooker for 2 min. The primary antibody anti-CYR61/CCN1 rabbit polyclonal (Abcam, Cambridge, UK) was used at a concentration of 1:250. A secondary antibody (anti-rabbit IgG HRP polymer (ready to use, Vector, 2BScientific, Upper Heyford, UK)) was used for signal amplification and DAB (Abcam, Cambridge, UK) reaction (10 min) was used for signal detection. Slide analysis was carried out by calculating and combining two scores for the CCN1 staining as described above for TMA analysis. The samples were randomised prior to scoring to prevent bias.

### 2.4. Enzyme-Linked Immunosorbent Assay (ELISA)

GBM patient blood samples matched to tumour samples were collected, processed and stored by the Leeds Multidisciplinary Routine Tissue Banking (RTB) service from GBM patients undergoing surgery at the Leeds General Infirmary (ethical approval no. RTB 15/YH/0080). A commercially available Human CYR61/CCN1 Quantikine ELISA kit (R&D Systems, Abingdon, UK) was used for the solid-phase sandwich ELISA for CCN1. The ELISA was carried out following the manufacturer’s instructions, using 100 µL of assay diluent per well, followed by 50 µL of patient blood serum in duplicate, control blood serum (from healthy donors), or standard supplied in the kit. After the final incubation step the plate was immediately read out on a plate reader at 540 nm (ThermoFisher, Altrincham, UK). 

For determination of CCN1 secretion from established cell lines, the same ELISA kit was used. U251 and U87 cells were split to a density of 2 × 10^3^ into 6-well plates (Costar, Corning Lifesciences, New York, NY, USA) and incubated for 24 h at 37 °C. Following incubation, the media was removed and replaced with 2 mL of complete medium (control + DMSO only) or medium with 500 nM of CCG-1423 (originally resuspended in DMSO) (Tocris, Bristol, UK). The cell supernatants were collected at 24, 48 and 72-h time points. For the ELISA, three wells per time point per cell line and 8 for the standard curve were set up and 100 μL of assay diluent was added to each well, followed by 50 μL of cell supernatant, control or standard. The same protocol as for the blood samples was followed.

### 2.5. Live Cell Imaging

U251 and U87 cells were adjusted to a density of 1.5 × 10^3^/mL in complete medium. 100 μL of either cell suspension was pipetted into each well of a flat bottomed 96 well plate (half a plate per cell line). Cells were allowed to settle for 24 h at 37 °C (humidified atmosphere with 5% CO_2_). For the experiments, 100 μL of complete medium containing either DMSO (control) or inhibitor diluted in DMSO (20 mM lithium chloride (LiCl), 5 µM 6-bromo-indirubin-3′-oxime (BIO), Selleckchem, Cambridge, UK), 500 nM CCG-1423, all diluted in DMSO) was added. Live cell time-lapse imaging using the IncuCyte Zoom System (Essen BioScience, Royston, UK) started immediately after drug application for 72 h at 37 °C (humidified atmosphere with 5% CO_2_). Movies were generated using the IncuCyte Zoom System software. For analysis and quantification, cell motility was tracked and measured with the MTrackJ plugin for ImageJ ( Rasband, W.S., ImageJ, U.S. National Institutes of Health, Bethesda, MA, USA, https://imagej.nih.gov/ij/, 1997–2018, accessed on 14 December 2021). Migration distance was determined as the accumulated total migration distance over the given time. Displacement as the length of the vector from start to end-point of measured migration was calculated to give an indication of directionality of movement.

### 2.6. Immunofluorescence (IF) and Analysis of MKL1 Intracellular Distribution

U251 cells were cultured from a density of 2 × 10^3^ on sterile 25 × 25 coverslips in 6 well plates (Costar, Corning Lifesciences, New York, NY, USA) allowed to settle for 24 h at 37 °C. Cell culture media was replaced with 2 ml of complete medium with DMSO (control) or medium with 500 nM of CCG-1423 (originally resuspended in DMSO, Tocris, Bristol, UK). Cells were incubated for a further 48 h with control/inhibitor medium at 37 °C. The cells were then fixed using 4% PFA (Thermo Scientific, Altrincham, UK) and blocked for 1 h using 5% normal goat serum (Abcam, Cambridge, UK). The cells were then labelled for MKL1 (anti-MKL1, made in Rb, Abcam, 1:100 in blocking solution, Cambridge, UK) and incubated for 24 h at 4 °C. After 3 PBS washes and incubation for 1 h with the secondary antibody (anti-rabbit AlexaFluor 488 (Abcam, Cambridge, UK), 1:500) the cells were washed again 3 times in PBS and mounted on glass microslides using Mowiol containing DAPI nucleus stain (Sigma, Poole, Dorset, UK). Samples were imaged on a Zeiss LSM880 inverted confocal microscope (Zeiss, Birmingham, UK) and resulting images were analysed in ImageJ, as follows: Threshold images were created using standard ImageJ settings for the DAPI and 488 channel separately. These were used to create ROIs outlining the nucleus (DAPI channel) or the respective (single) cell (488 channel). These ROIs were then used to calculate the mean grey values of MKL1 fluorescence in the cytoplasm (ROI cell minus nucleus) and nucleus (ROI nucleus). 

### 2.7. Western Blots 

Cell lysates were obtained from cells grown in 25 cm^2^ plastic tissue culture flasks (Corning) treated with 500 nM of CCG-1423 (Tocris, Bristol, UK) for 24, 48 and 72 h, respectively. Cell culture vessels were placed on ice and cell cultures were washed twice with ice-cold PBS. Cells were collected and pelleted at 400× *g* for 5 min. PBS was removed and the cell pellet was resuspended in 0.5 mL of Tris-HCl lysis buffer, containing 25 µL/mL protease inhibitor (Sigma, Poole, Dorset, UK) and incubated on ice for 5 min. For cytoplasmic separation, the solution was centrifuged at 400× *g* for 10 min. Supernatants containing the cytoplasmic extract were collected and transferred into clean tubes. The protein concentration of each cell lysate was determined by Bradford protein assay (BioRad, Watford, UK). For Western blotting, proteins were separated by SDS-PAGE and transferred to PVDF membranes (BioRad, Watford, UK). Antibodies used for immuno labelling of Western blots were rabbit anti-MKL1 (1:300, Abcam, Cambridge, UK) and HRP-conjugated donkey anti-rabbit IgG (1:1000, GE Healthcare, Chalfont Saint Giles, UK). Proteins were detected by SuperSignal West Femto maximum sensitivity substrate (ThermoFisher, Altrincham, UK), according to the provided protocol and visualised using a ChemiDoc MP imaging system (BioRad, Watford, UK) and associated Image Lab software (BioRad, Watford, UK). Protein band intensity was analysed using ImageJ software and normalised to the corresponding β-actin control.

### 2.8. 3D Spheroid Generation and Invasion Assay

U87-NT cells were seeded at 3 × 10^3^/well in low adherence 96 well plates (ThermoFisher, Altrincham, UK) as previously described [14]. Three days after seeding, spheroids contained within the wells were embedded in rat tail collagen V (Corning Life Science, Glendale, AZ 85301, USA), polymerisation was achieved with 1 M NaOH. The inhibitors CCG-1423 and Rhosin-HCl (Tocris, Bristol, UK) were resuspended in DMSO and added at a predetermined anti-migratory concentration (CCG-1423: 500 nM, Rhosin: 1 µM). Control spheroids were mock-treated with DMSO-supplemented medium only. Spheroids were allowed to grow for 72 h in collagen, fixed with 4% PFA and labelled with Phalloidin-TRITC (ThermoFisher, Altrincham, UK) in their collagen plugs. Image z-stacks of spheroids were acquired on a Zeiss 880 LSM confocal microscope (Zeiss, Birmingham, UK) with an EC plan-neofluar 10× objective. For analysis maximum projections of the z-stacks were used. The resulting grey-scale images were inverted and using the free-hand selection tool in ImageJ, the outline of the spheroid as well as of the gaps between migration cells were drawn and the area measured (for illustration see Appendix A).

### 2.9. Data Analysis 

For all experiments, quantified data were statistically analysed using RStudio 1.4.1106 (RStudio Inc., Boston, MA, USA) using the R packages ggplot2, tidyverse, ggbeeswarm, rstatix, and ggpubr. Data were tested for normality using the Shapiro Wilk test. Accordingly, ANOVA with Tukey post hoc test or *t*-test for normally distributed data, otherwise Kruskal–Wallis with post hoc Dunn test, or Wilcoxon–Mann–Whitney-test were performed. Used scripts are available on request. *p*-values < 0.05 were considered statistically significant. 

## 3. Results

### 3.1. CCN1 Expression Correlates with Tumour Grades in Patient Tissues

To obtain an overview of CCN1 expression in different types and grades of brain tumours, we employed a commercially available TMA assay. In general, cytoplasmic, and membranous CCN1 staining was visible in tumour cells, with strong staining in elongated cells and cells in close proximity to blood vessels (data not shown), suggesting a correlation between CCN1 and invasive cell migration. The clinical characteristics associated with the samples, i.e., age, sex, and tumour grade, are summarised in a table in Figure 1A. Samples were dichotomised into low and high CCN1 expression with low scoring samples having a score of 3 or less and high scoring samples having a score of 4 or more. High CCN1 expression correlated with high astrocytoma grade (e.g., 50% astrocytoma grade 4 samples were scored 4 or higher), while low CCN1 expression was observed within low astrocytoma grades. Quantification and comparison of the different brain tumour types revealed a clear correlation of high CCN1 expression with GBM (tumour grade 4) that was significantly different from the negative control and low-grade astrocytoma (Figure 1B).

We further analysed CCN1 expression judged from IHC staining on a total of 16 GBM samples collected from 9 patients undergoing surgery. We followed the procedure and scoring for the TMA samples. Tumour edge samples, defined as 200 µm from the tumour boundary, and tumour core samples, defined as the tumour region outside the edge margins [18], were collected from 4 of the 9 patients, 1 of which also had a sample of cancer adjacent tissue collected. Two of the patients had an additional later surgery for recurrent tumours, from which samples were also collected. Figure 1C summarises the resulting scoring for these samples. In brief, all samples but one were scored as highly expressing CCN1 (score 4 or higher). Interestingly, where core and edge samples were available, the edge sample scored equal (patient 30) or higher than the core (patient 40, 52, 58, Figure 1C). Moreover, samples from recurrent samples always scored higher, than the primary sample. This trend also emerged in the matched blood samples, collected from each patient at the time of surgery (Figure 1D). These matched samples allowed us to investigate a possible correlation between CCN1 levels in the blood of cancer patients and tumour aggressiveness/severity. CCN1 is known to be associated with angiogenesis and we did observe distinct CCN1 labelling around vessels in TMA samples. Thus, we assumed CCN1 blood levels may be detectable if CCN1 was able to cross the blood–brain barrier, as the blood–brain barrier is known to be compromised during GBM formation and progression. Indeed, differences in CCN1 concentration in serum samples, compared to the control, were seen in four patients. As mentioned previously, the CCN1 concentration of two patients became especially elevated at the time of recurrent tumours (patients 30 and 63, Figure 1D). Comparing the IHC staining in tissue samples from patient 48, with the highest measured CCN1 blood concentration, to patient 52 with the lowest concentration, showed a low intensity of CCN1 immunostaining in the latter, but strong staining in patient 48 tissue.

These findings highlight a correlation between CCN1 staining intensity and concentration of CCN1 in the serum of patients. Moreover, the results suggest a correlation between tumour grade and CCN1 concentration in tissue and blood of patients. This opens the interesting opportunity that CCN1 may be a useful liquid biopsy marker for GBM diagnosis. 

### 3.2. Effects of CCG-1423 on Cell Migration in Live Cell Imaging 

To better understand the relevance of our findings on the correlation between CCN1 expression and tumour aggressiveness/grade and concentration in matched patient serum, we next investigated the effects of blocking CCN1 in the established glioma cell lines U251 and U87. CCG-1423 is known to inhibit CCN1 production [13] and was therefore used to observe effects on cell migratory behaviour by live-cell imaging. The commercially available inhibitors BIO and Lithium chloride (LiCl) are well known for their inhibitory effects on migration in glioma cells [19,20] and were used as positive controls for comparison with CCG-1423 effects. Migratory distance (i.e., accumulated travelled distance over time), and cell displacement (i.e., vector length between cell positions at start and end of experiment) of treated cells are shown in Figure 2A,B and Appendix A.

After treatment with BIO or LiCl, migration (both in terms of distance and displacement) was significantly reduced in both cell lines in comparison to the control, as one would expect. The effects of BIO were observed almost immediately, within 10 min of treatment, in both cell lines and were maintained throughout the 72 h of imaging. Although the effect of LiCl U87 appeared to wane around the 48 h mark, the cells started to recover and adopt their normal cell migratory behaviour. In comparison, U251 cells did not recover for the duration of the imaging. Unlike the other inhibitors, the CCG-1423 treatment did neither result in changes in migration distance nor displacement, compared to the control cells. These results were surprising in the light of the previously reported data suggesting a significant decrease in PC3 cell invasion when treated with CCG-1423 [13].

To ascertain further the impact of treatment of CCG-1423 in blocking CCN1 production, we investigated the levels of secretion of CCN1 pre- and post-application of CCG-1423. As CCN1 is secreted by cells, a sandwich ELISA was used to detect the presence of CCN1 in the supernatant cell culture medium of CCG-1423-treated and untreated U251 and U87 cell cultures over a time span of 72 h. We found, indeed that the concentration of CCN1 in the medium did not increase in treated cells, whereas CCN1 levels clearly increased in supernatants from untreated cells over time (Figure 2C,D). From these results, it was evident that CCG-1423 had an effect on the secretion and/or production of CCN1; despite cells continuing to migrate, as illustrated by the live-cell imaging results. This suggested a previously unreported relationship between CCG-1423 exposure and CCN1 expression.

### 3.3. Loss of Nuclear Import of MKL-1 Is Facilitated by CCG-1423 Activity

To investigate this further, we decided to investigate the regulation of CCN1 production. CCG-1423 binds to the transcriptional co-activator MKL1, preventing its nuclear import and therefore activation of MKL1/SRF-dependent transcription and CCN1 production [15]. Immunofluorescence labelling was used to visualise the accumulation of MKL1 in the cytoplasm of U251 cells treated with CCG-1423 in comparison to untreated control cells. As U87 and U251 cell lines had shown so far equal reactions to CCG-1423 treatment and U251 is the more representative cell line in terms of adopting mesenchymal migration with a pronounced lamellipodia front as observed by live-cell imaging, only results for U251 is shown here. There appeared to be a marked increase in cytoplasmic MKL1 in cells treated for 72 h with CCG-1423 with a concomitant loss of nuclear MKL1 in comparison to the control (Figure 3A,B).

To confirm the results from the IF imaging of MKL1, Western blots were carried out to determine cytoplasmic protein levels of MKL1 in untreated control and CCG-1423 treated U251 cells over a 72 h period. During sample preparation of cell lysates, nuclear-associated components were separated and discarded (as described in materials and methods), therefore the results from the Western blot analysis represent solely cytoplasmic MKL1. This biochemical analysis of U251 cells showed an increase in MKL1 accumulation in the cytoplasm when treated with CCG-1423 (Figure 3C,D), supporting the IF results. The signal intensity of MKL1 was clearly stronger (after normalisation with actin) in CCG-1423 treated U251 cells at 24 and 48 h time points. After 72 h treatment, signal strengths converged, which is consistent with the live-cell imaging results, showing that cells returned to normal behaviour after 72 h CCG-1423 treatment. Western blot analysis for cytoplasmic MKL1 in untreated control versus CCG-1423-treated cells was further carried out on the cell lines HOG and G44 grown as neurospheres, the murine glioma cell line GL261 and rat glioma cell line CNS1 (Appendix A). As for the U251 cell line, all additional cell lines showed clearly elevated signals for the intensity of the MKL1 band in CCG-1423 treated cells compared to the control at all analysed time points (Appendix A). These results, along with the MKL1 immunofluorescence analysis of cellular localisation and CCN1 ELISA, illustrate that CCG-1423 can prevent CCN1 production through inhibition of the nuclear import of MKL1. 

### 3.4. Targeting RhoA Signalling and Rac Activation Reduces Cell Migration by a Switch to Collective Cell Migration

In our previous publication [15], we reported on a mesenchymal–amoeboid switch in the U251 cell line characterised by loss of major protrusions/lamellipodia in cells treated with CCG-1423 and a striking increase in the number of filopodia, shown here at high resolution to highlight the phenotypic differences (Figure 4A). Following on these original findings, we reasoned that targeting Rac activation and RhoA-driven cell migration by a combination treatment with CCG-1423 and a RhoA inhibitor, such as Rhosin HCL, will reduce cell migration in the cells. We used a combination treatment and then analysed U87-generated spheroids to assess the effect of the combination treatment by confocal microscopy. We utilised U87 as a proof of principle here as U87 rapidly migrates in a 3D, collagen-based environment characterised by producing extensive protrusions emanating from the original spheroid. Our preliminary findings indicate that treatment of the cells with a combination of CCG-1423 and Rhosin HCl induced a third shift potentially towards a collective cell migration as indicated by our findings that gap areas within the protrusions were significantly reduced (Figure 4B,C).

## 4. Discussion

CCN1 is known to be overexpressed in a number of different cancer types and to drive invasion [21,22,23,24]. Elevated levels of CCN1 in laryngeal squamous cell cancer promoted the induction of epithelial–mesenchymal transition (EMT), thus leading to invasion and metastasis and resulting in a poor prognosis [25]. In addition, CCN1 (as well as other CCN family members) has been shown to be dysregulated in colorectal cancer and to be involved in the initiation and development as well as the promotion of this disease [26]. Furthermore, a study in osteosarcoma tumours determined that silencing CCN1 reduces tumour vascularisation and slows the growth of osteosarcoma cells resulting in a reduction in subsequent lung metastases [27]. These few examples indicate towards diverse actions of CCN1 on cancer progression.

In the current study, CCN1 levels were explored in patient-derived glioma tissue and GBM cell lines. CCN1 levels were elevated both in its secreted form in blood samples and in tumour tissues from glioma patients. In tissues, CCN1 was expressed in the cytoplasm of tumour cells, with strongly stained cells associated with blood vessels. Moreover, CCN1 expression levels were concomitant with high-grade tumours, especially in comparison to expression levels in lower-grade gliomas. This data suggests that CCN1 expression is correlated with tumour aggressiveness, which confirms a previous study that performed a semi-quantitative IHC analysis of gliomas and normal brain samples correlating CCN1 with tumour grade: In 88% of the WHO grade IV samples studied, CCN1 was overexpressed, and expression was positively correlated with the expression of c-Met, a receptor tyrosine kinase involved in proliferation, migration and invasion. When investigated in vivo using a U87 xenograft in mouse models, CCN1 siRNAs significantly inhibited proliferation by 57% compared to the control [11].

Interestingly, in our study levels of secreted CCN1 in the GBM tissue, samples were mirrored in the matched patient blood in comparison to blood samples from healthy volunteers. In addition, samples from patients with recurrent tumours showed a significant increase in CCN1 levels both in tumour tissues as well as in blood samples in comparison to the primary tumour and blood sample. As the observed increases in CCN1 levels are considerable, the use of larger sample sizes and further investigations could prove CCN1 to be a potential blood biomarker for the monitoring of brain tumour patients to predict disease recurrence and help adapt further treatment.

Moreover, improved sensitivity to enhance the detection levels of an ELISA-based screening system may lead to the development of a cost-effective and non-invasive diagnostic tool for advanced early detection of brain tumours, an unmet need in glioma. The molecular subtyping of brain tumours themselves has advanced diagnosis and treatment options; however, a conclusive diagnosis is still largely based on the interpretation of the histopathology of tumour samples, which may be open to human error and bias. The growing identification and use of biomarkers and genomics data will enhance brain tumour diagnosis by providing more detailed information when paired with the traditional histopathology method [28].

The distribution of CCN1 identified in our clinical sample derived data provided the basis for our functional studies. Previous investigations revealed that the small molecule inhibitor CCG-1423 inhibits the production of CCN1, followed by a reduction in cell migration, [11,12]. Unexpectedly, our findings showed seemingly unaffected migration despite CCG-1423 treatment. However, indeed, we observed a striking relocalisation of the co-transcriptional activator MKL1 from a nuclear to a cytoplasmic localisation as a result of CCG-1423 treatment, e.g., in U251 cells. Western blot analysis of U251 cell lysates also uncovered a build-up of MKL1 in the cytoplasm of cells treated with CCG-1423 at 24 and 48 h time points with cells appearing to return to pre-treatment balance after 72 h. This is consistent with the original CCG-1423 inhibition studies conducted by Evelyn et al. (2007) on PC-3 cells, whereby cells recovered following the withdrawal of CCG-1423 [13]. Furthermore, we identified a significant decrease in CCN1 secretion in the supernatants of U251 and U87 cells treated with CCG-1423, confirming the action of the inhibitor. However, there was no apparent overall effect on cell migration in cells treated with CCG-1423, judged from our migration studies. In our previous study, using iSIM imaging and quantification of cell extensions, we showed a direct effect on the frequency of filopodia and major cell protrusions of U251 cells due to incubation with CCG-1423 [15]. We included additional representative iSIM images from this data set, a quantification of the data was reported in this previous publication. U251 cells reacted to the inhibitor with a significant reduction in major protrusions and adopted a more rounded shape, which indicates a transition from mesenchymal to amoeboid migration [29]. These results led us to hypothesise a potential induction of MAT due to CCN1 inhibition as a drug resistance mechanism adopted by glioma cells. 

Previous studies suggest that CCN1 secretion results in inhibition of Rac activation leading to reduced mesenchymal migration and loss of focal adhesions with a concomitant increase in rounded cell movement [30]. The extensions of the plasma membrane in lamellipodia of cells are primarily driven by Rac-activated actin polymerisation. An essential part of migration involving lamellipodia is integrin-mediated adhesion, as this maintains activated Rac in a positive feedback loop whereby integrins at the leading edge of a cell stimulate Rac activation [31]. Rac recruits activated integrins to the leading edge to promote cell migration [32]. Among these is the integrin αvβ3, whose expression is in turn stimulated by CCN1 [33]. By inhibiting CCN1, integrin αvβ3 will not be activated and therefore, will not be recruited by Rac to stimulate the positive feedback loop and promote elongated cell migration. Because of this reduced Rac activation, rounded motility of cells will be elevated via a bleb-driven cell migration, such as amoeboid migration. This motility type correlates with a high level of active RhoA/ROCK signalling and is driven by cortical actomyosin contractility [34]. This migratory activity is often seen in vivo and in low-adhesion in vitro systems. Cells have the ability to readily transition between bleb-based migration (amoeboid) and lamellipodia-based migration (mesenchymal) in vivo in order to adapt to their surroundings as they migrate [35].

CCG-1423 activity targets events downstream of the RhoA signalling pathway but upstream of CCN1 production and therefore Rac activation. Not only does this explain the reduction in CCN1 cell secretion but also the changes in phenotype from elongated to rounded. This correlates with the concept that CCG-1423 is inducing MAT in these cells in vitro, allowing them to continue to migrate after treatment with CCG-1423 by amoeboid migration. We were able to observe a distinct loss of major protrusions/lamellipodia in treated cells compared to the control cells, which was concurrent with an increase in the number of filopodia in treated cells compared to control cells, a striking feature of MAT (Figure 4A). These results support a direct effect on actin dynamics and cell morphology indicative of MAT when cells are treated with CCG-1423. Anti-migratory combination treatments to prevent tumour cells from migrating to healthy parts of the brain will need to be developed to circumvent a potential migratory switch from one phenotype to another. Finally, based on findings by Butler [36], that a combination treatment of U87 with CCG-1423 and an inhibitor targeting RhoA-driven migration reduced cell migration in 2D and 3D, we investigated this effect by confocal microscopy. In a preliminary experiment we treated U87 tumour spheroids with either CCG-1423, or the RhoA inhibitor Rhosin HCl, or a combination of both inhibitors, which revealed a strong effect of both inhibitors in combination on cell invasion into collagen. In addition, by confocal microscopy, for the first time, we were able to uncover that the combination of both inhibitors leads to a more sheet-like collective migration compared to the loose single-cell migration in control or single inhibitor groups. Cancer cells have been reported to adopt this type of cell migration highlighting great plasticity in migratory patterns [37]. This apparent ability of glioma cells to respond to extracellular challenges and adopt to continuous cell migration and invasion requires further detailed investigation including additional in vitro methods to characterise RhoA activation and MMP activity as reported by Chikina and Alexandrova (2018) [38]. In addition, the downstream events regulated by MKL1 activity should also be further investigated.

## 5. Conclusions

The studies presented here provide novel and compelling evidence that glioma cells can undergo MAT in response to pharmacological intervention intended to prevent migration, resulting in continued migration and invasion, which supports the notion of combination treatments with migrastatic inhibitors. This also highlighted the importance of moving cell migration research into a 3D setting to obtain more accurate drug efficacy results in a more tumour-relevant environment. In addition, our study has uncovered an important role for CCN1 in disease progression supporting CCN1 as a potential biomarker for GBM, which could help disease monitoring and improve survival rates of this devastating disease.

## Figures and Tables

**Figure 1 biomedicines-10-00009-f001:**
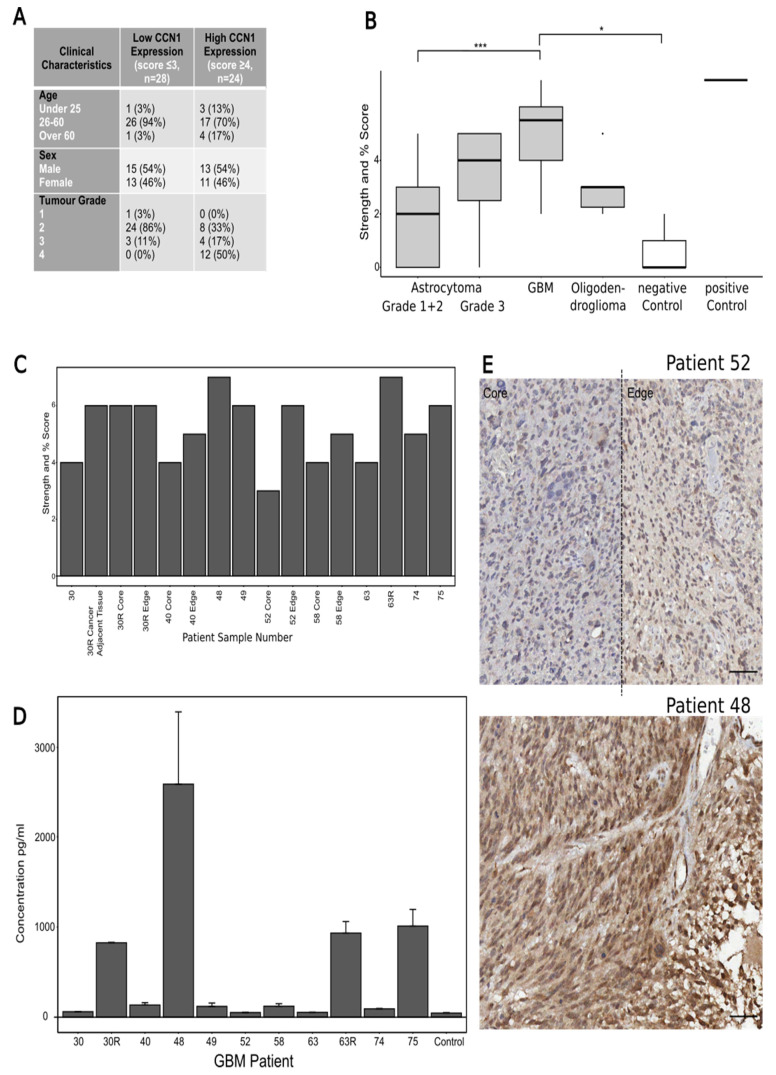
**CCN1 expression in patient samples.** (**A**) Combined scoring of CCN1 staining strength and percentage of the core covered on a TMA containing astrocytomas grade 1–2, GBM, oligodendroglioma, cancer adjacent brain tissue as a control and an adrenal core as a positive control. (**B**) The clinical characteristics of the TMA astrocytoma samples. The results reveal associations of CCN1 expression levels with tumour grade. 86 % of the samples with low CCN1 expression are grade 2 astrocytomas, whilst 50% of the samples with high CCN1 expression are grade 4 astrocytomas (GBM). There were no associations of CCN1 levels with age or sex. (**C**) Combined scores of CCN1 staining intensity and percentage of staining covering the samples collected from 9 GBM patients during surgery. Recurrent tumour samples were also collected for patients 30 and 63 (30R and 63R). Samples of the core and edge of the tumour were collected for patients 30, 40, 52 and 58. Patient 30 also had a sample of cancer adjacent tissue collected. (**D**) CCN1 concentration in 9 patient blood serum samples. Blood samples from patient no. 30 and 63 were also collected at their recurrent GBM tumour surgery (30R and 63R). The percentage increase of each patient compared to the control was also calculated. (**E**) CCN1 IHC staining correlates with the CCN1 concentration determined for the matched blood samples. The GBM tissue sample from patient no. 52 presented with the lowest concentration of CCN1 in the blood (51.51 pg/mL) which correlates with the low intensity CCN1 staining in the tissue samples. The GBM tissue sample from patient no. 48 presented with the highest concentration of CCN1 in the blood (2588.041 pg/mL), which correlates with the high intensity CCN1 staining in the tissue sample. Scale bar = 50µm, Magnification 20×. Asterisks show statistical significance. * *p* ≤ 0.05, *** *p* ≤ 0.001, Kruskal–Wallis: *p* = 0.0000344. Significant differences were marked in the graphs when available.

**Figure 2 biomedicines-10-00009-f002:**
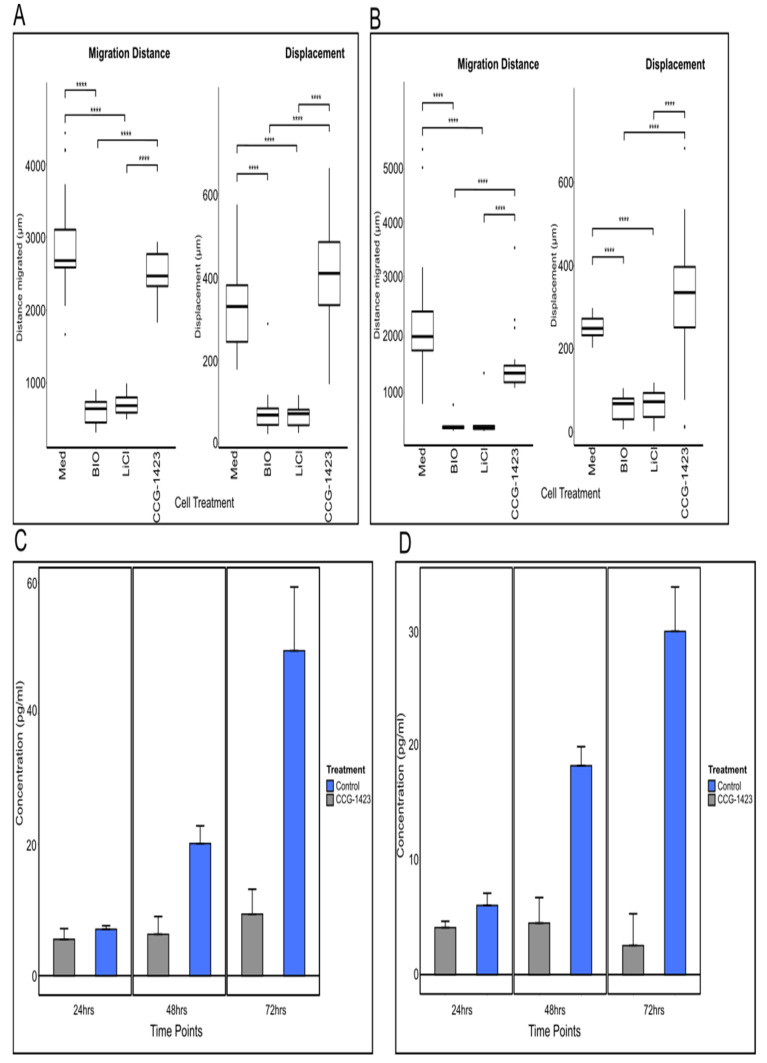
**Effect of CCG-1423 on cell migration and CCNI secretion.** The effects of inhibitors on cell migration as measured by distance travelled and cell displacement in (**A**) U251 and (**B**) U87 cell lines. Experiments were repeated in triplicate. For sample specific tracking examples see Appendix A. Asterisks indicate statistically significant results compared to the control (Med). **** *p* ≤ 0.0001, Kruskal–Wallis for A, Distance: *p* = 4.18 × 10^−13^, Displacement: *p* = 1.52 × 10^−12^; for B, Distance: *p* = 6.62 × 10^−13^, Displacement: *p* = 2.17 × 10^−10^. An ELISA was carried out to determine CCN1 levels in the supernatants of (**C**) U251 and (**D**) U87 cell lines over a 72-h period. Cells were incubated for 24, 48 and 72 h +/− CCG-1423. The supernatants were collected for each time point and tested for CCN1 by ELISA. The results suggest reduced CCN1 concentration levels in the supernatants of treated U251 and U87 cells, at the 48 and 72-h time points, when compared to the untreated controls as indicated by decreased concentration. Significant differences were marked in the graphs when available. **** *p* ≤ 0.0001.

**Figure 3 biomedicines-10-00009-f003:**
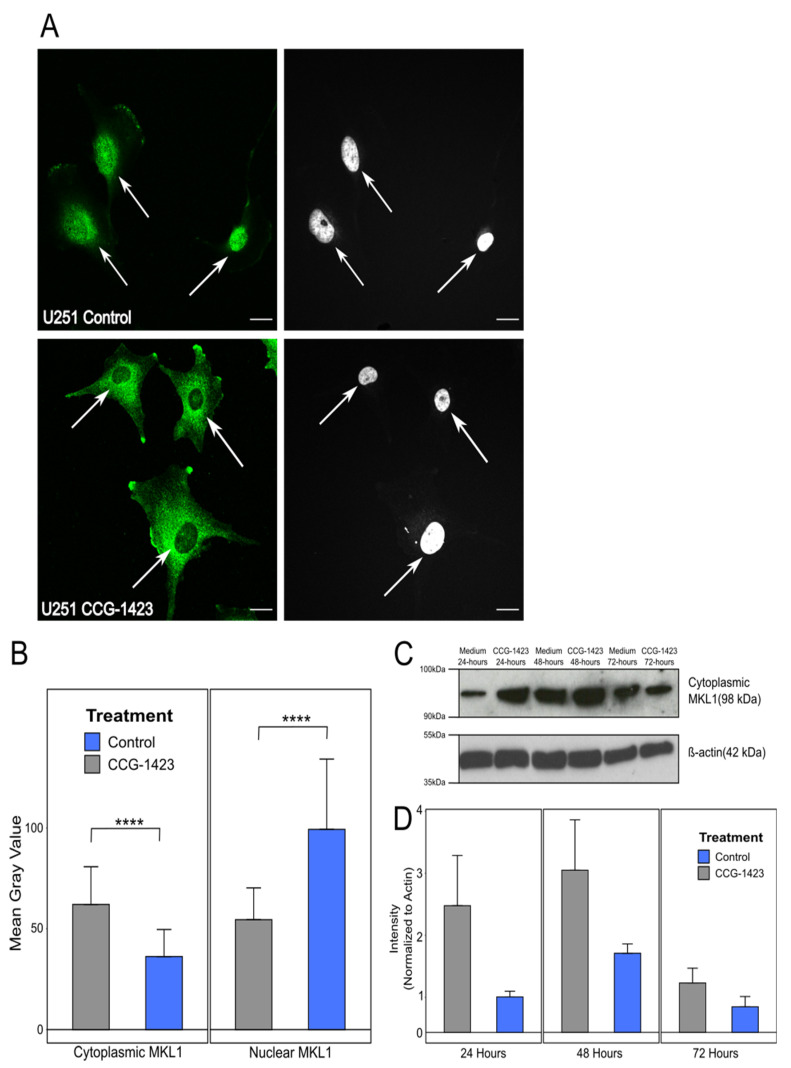
**Immunofluorescence staining and analysis of MKL1 in untreated and CCG-1423 treated U251 cells.** (**A**) Arrows show a build-up of MKL1 in the cytoplasm of treated cells compared to the control cells which show pronounced nuclear staining. Imaged on a confocal microscope at a magnification of x63. Green fluorescent stain = MKL1; right panel: nuclear DAPI stain; Scale bar = 20 µm. (**B**) Quantification of immunofluorescence staining of MKL1 to determine differences in nuclear and cytoplasmic MKL1 localisation in untreated (control) U251 cells and U251 cells treated with CCG-1423. The images were analysed using ImageJ where a mean gray value was calculated for the nucleus and the cytoplasm of each cell. Kruskal–Wallis: *p* = 1.96 × 10^−31^ (**C**) The U251 cell line was treated with 500 nM of CCG-1423 and incubated for 24, 48 and 72 h before cell lysates were obtained. Protein levels of cytoplasmic MKL1 were then determined by western blot. A beta-actin control was included as a protein loading and transfer control. (**D**) Western blot analysis of the effects of CCG-1423 on U251 cells. MKL1 band intensity was quantified and normalised to the β-actin control using Image J software. Graphs show mean ± SD of 3 individual experiments. Asterisks show statistical significance where **** *p* ≤ 0.0001. Significant differences were marked in the graphs when available. **** *p* ≤ 0.0001.

**Figure 4 biomedicines-10-00009-f004:**
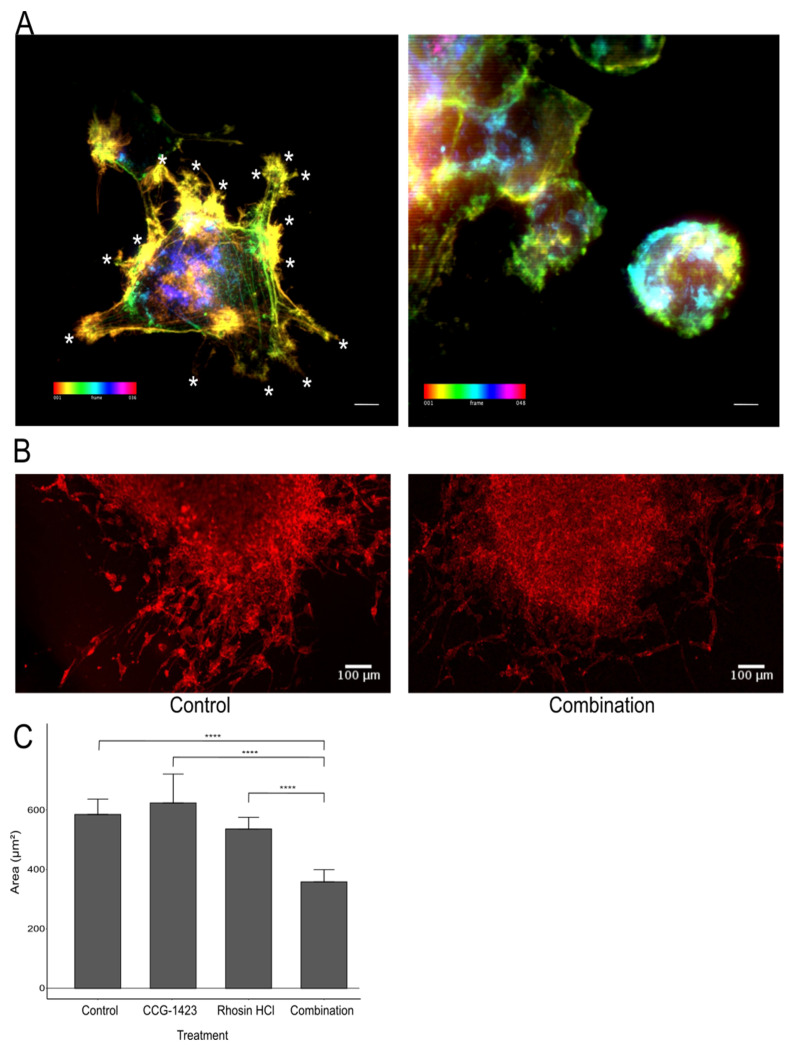
**High resolution iSIM imaging and analysis of fixed spheroids and migrating cells within collagen plugs.** (**A**) Top image: a single cell in untreated spheroids emanating from the original spheroid core. Asterisks show filopodia. Right hand image: the elongated shape disappears after treatment with inhibitor and is replaced by a rounded phenotype. Fluorescent label = Alexa Fluor 488 phalloidin. Colour bar: sample depth. Scale bar = 10 µm. (**B**) Confocal microscopy visualizes the effect of drug treatment on U87 spheroids. Images of fixed U87 treated spheroids, in their collagen plugs, taken on a Zeiss LSM 880 with an EC Plan-Neofluar 10× objective lens. Cells express GFP (488 nm) and are labelled with TRITC conjugated phalloidin (594 nm), to highlight the actin cytoskeleton. Migration appears more sheet-like in response to the combination treatment, in comparison to the mesenchymal and amoeboid extensions seen migrating from the control spheroid. Scale bar represents 100 µm. (**C**) Average U87 collagen gap area. U87 cells were treated with CCG-1423 (500 nM), Rhosin HCl (1 µM) or a combination of both. The combination of both inhibitors show a significant (*p* < 0.0001) difference to untreated (Control) and all single inhibitor treatments (Kruskal–Wallis *p* = 3.08 × 10^−10^). See Appendix A for quantification illustration. Significant differences were marked in the graphs when available. **** *p* ≤ 0.0001.

## Data Availability

The data presented in this study are available on request from the corresponding author.

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
