# Peer review of "Drug Resistance in Glioma Cells Induced by a Mesenchymal–Amoeboid Migratory Switch"

_biomedicines, 2021, doi:10.3390/biomedicines10010009_

Round 1

Reviewer 1 Report

The current manuscript is an interesting study reporting that even CCG-1423 treatment decreases CCN1 secretion, it did not prevent overall glioma cell migration. The authors found that induction of MAT is a drug resistance mechanism that leads to glioma cells continue to migrate. However, there are also several issues the authors should address as summarized below.

  1. Although the authors found a new mechanism why glioma cells continue to migrate after CCG-1423 treatment, they still did not report a method to prevent glioma cell migration. This is very necessary to make this manuscript interesting to readers.
  2. The authors showed the high expression of CCN1 in patient tissues in Figure 1. How about the CCN1 expression in all the tumor cell lines used in this study (U251, U87, G9, GL261, etc)?
  3. The authors need to add a title in each figure in Figure 2. The authors should also indicate in the main context where is the description of Figure 2C and 2D.
  4. Resolution of all the figures needs to be improved.

Author Response

The current manuscript is an interesting study reporting that even CCG-1423 treatment decreases CCN1 secretion, it did not prevent overall glioma cell migration. The authors found that induction of MAT is a drug resistance mechanism that leads to glioma cells continue to migrate. However, there are also several issues the authors should address as summarized below.

  1. Although the authors found a new mechanism why glioma cells continue to migrate after CCG-1423 treatment, they still did not report a method to prevent glioma cell migration. This is very necessary to make this manuscript interesting to readers.
  2. The authors showed the high expression of CCN1 in patient tissues in Figure 1. How about the CCN1 expression in all the tumor cell lines used in this study (U251, U87, G9, GL261, etc)?
  3. The authors need to add a title in each figure in Figure 2. The authors should also indicate in the main context where is the description of Figure 2C and 2D.
  4. Resolution of all the figures needs to be improved.

Response: We would like to thank the reviewer for their interest in our study. We would like to respond to the reviewer’s comments:

  1. This is certainly an important point. Since this work was completed, we were able to continue with some studies at the University of Huddersfield. In our follow-on studies by one of our students we were able to show that a combination treatment of U87 with the RhoA inhibitor Rhosin hydrochloride did show a synergistic effect on cell migration (this inhibitor did not have an effect on its own) indicating that combination treatment targeting the two different migration signalling pathways does induce anti-migratory effects. We have included this now in the text (see Butler: Butler, H., (2021) “Combination treatment with migrastatic inhibitors to target brain tumour spread”, Fields: journal of Huddersfield student research7(1). doi: https://doi.org/10.5920/fields.823 also in the text line 438.
  2. At the time of the study the ELISA kit was only available for the two cell lines we started off with, U87 and U251, therefore we do not yet have the data for the other cell lines; this is certainly a good point to start on for follow on studies.
  3. We have now corrected this in the figure legend, see line. Description of figure 2c and d has been included. See in the manuscript, figure 2.
  4. All figures have been improved in terms of resolution.

Reviewer 2 Report

A brief summary:

Motility and invasion are key hallmarks of glioblastoma (GBM) aggressiveness. Understanding the mechanisms of migration may improve therapeutic interventions to inhibit invasiveness by GBM cells.  The authors of this manuscripts investigated the drug resistance mechanism induced by CCG-1423, a CCN1 inhibitor. First, CCN1 expression levels were associated with high grade GBM tumors in comparison to lower grade gliomas, finding also a significant increase in the blood samples of recurrent GBM patients. Then, CCN1 secretion and motility after CCG-1423 treatment was evaluated in two GBM cell lines, and MKL1 cytoplasmic accumulation was also reported. A final study evaluating filopodia and protrusion was performed to confirm the mesenchymal-amoeboid transition induction after CCG-1423 treatment.

Broad comments:

Quality of presentation is poor and most of results are not written properly. Data and analyses are not presented correctly. There are too many flaws that do not allow a smooth reading and a clear understanding of the results. Resolution of all figures is weak. They should be improved. All figures have a long and complex description of the caption; it is not easy to read and understand the results because a lot of information, including materials and methods are mixed with the caption. Furthermore, most of the results do not show a correct statistical comparison of the experimental groups; the comparison is two-tailed T-test between two groups, but ANOVA among more experimental groups is required for a correct interpretation of the results, especially when they are represented in the same graph (i.e., among time points). Above all these flaws, the crucial point of the research is about the demonstration that CCG-1423 reduced CCN1 levels, but GBM cells continued to migrate. The question is that the migration results present many critical issues since the methodology is not explained correctly and representative images are not shown to confirm the results. Hence, the fundamental point of the study is not properly addressed. In conclusion, high resolution by Instant Structured Illumination Microscopy presents lacks in results and methodologies for mesenchymal-amoeboid transition, that is a second key point of the manuscript; but this process was not evaluated properly and no additional assays were performed to confirm mesenchymal-amoeboid transition in GBM after CCG-1423 treatment. Such a flaws prevents any conclusion to be made.

Specific comments:

  1. In page 5 line 16, check if Figure 1A referred to CCN1 staining is correct. Caption of Figure 1 is very long and complex. I would suggest reducing the description adding more information in the results and materials and methods section. CCN1 should be added in axis y of figures 1.
  1. Figure 2 has several flaws. Title is indicated as “Effects of CCG-1423 on CCN1 secretion”. This is not appropriate and coherent with the experimental activities and data showed regarding migration and morphological changes. The authors should also explain how the displacement and migration analysis is performed. There are no representative images to confirm the effects of migration, displacement and morphological changes. Time points in A and B are not showed. “Concentration of CCN1” should be added in axis y of figures 2c and d. Variation on CCN1 levels in the supernatants over time in figures 2c and 2d should be tested with statistical analysis among time points. The description of these results is not clear.
  1. In page 8 lines 19 the reference to Figure 2B is not correct.
  1. It is necessary the nuclear counterstaining in Figures 3 and merged images with nuclei. Moreover, three cells per ROI are not representative to confirm the results.
  1. Nuclear and cytoplasmatic MKL1 is compared in controls versus treated. It should be appropriate to compare also MKL1 Nuclear versus MKL1 cytoplasmatic in controls and MKL1 Nuclear versus MKL1 cytoplasmatic in treated. In western blot analysis MKL1 accumulation in the cytoplasm should be also validated comparing samples among 24 hrs, 48 hrs and 72 hrs (i.e. MKL1 band intensity of CGC-1423 over time; the same for control group). This point regards also supplementary figure 1.
  1. In page 9, lines 17-20. It is not clear the meaning, the interpretation and the reference about the results on live cell imaging.
  1. The authors wrote “The observed cytoplasmic build-up of MKL1 suggests the induction of MAT in glioma cells after treatment with CCG-1423”. According the previously results, the formulation of this hypothesis is not clear. Indeed, the authors showed that MKL1 accumulation on cytoplasm was associated to the CCG-1423 treatment. They also reported MKL1stimulates migration by CCN1 production after its nuclear translocation. Therefore, why should the accumulation of MLK1 provide an evidence of MAT induction?
  1. Figure 4 show a single cell. This is not representative to confirm the results. Moreover, color bar is not visible. Staining with Alexa Fluor 488 phalloidin is not reported in materials and methods.
  1. The effects of inhibitors on cell migration and CCN1 secretion was evaluated in U251-MG and U87-MG cells lines. Why do the cytoplasmic MKL1 localization and MAT induction were evaluated only in U251-MG cell lines?

Minor issues:

  1. English language is appropriate and understandable. I suggest doing minor check of English. Some sentences have an incorrect syntax. The conjunctions or punctuation are missing and the sentences are not linked correctly. Reading is not fluent in two examples below:

Page 1, Lines 40-43: “Nearly all patients will develop a recurrent tumour, despite therapy, and treatment options are limited to palliative approaches”.

Page 2, Lines 7-9: “In addition, in vivo in immunodeficient mice implanted with, for example, human gastric adenocarcinoma cells, overexpression of CCN1 is an effective inducer of angiogenesis, concomitant with tumour vascularisation and growth”

  1. A long list of P values ​​is found within the main result text. I suggest removing them all since the statistical analysis is shown in figures and their presence in the text confuses the reading.

Author Response

Quality of presentation is poor and most of results are not written properly. Data and analyses are not presented correctly. There are too many flaws that do not allow a smooth reading and a clear understanding of the results. Resolution of all figures is weak. They should be improved. All figures have a long and complex description of the caption; it is not easy to read and understand the results because a lot of information, including materials and methods are mixed with the caption. Furthermore, most of the results do not show a correct statistical comparison of the experimental groups; the comparison is two-tailed T-test between two groups, but ANOVA among more experimental groups is required for a correct interpretation of the results, especially when they are represented in the same graph (i.e., among time points). Above all these flaws, the crucial point of the research is about the demonstration that CCG-1423 reduced CCN1 levels, but GBM cells continued to migrate. The question is that the migration results present many critical issues since the methodology is not explained correctly and representative images are not shown to confirm the results. Hence, the fundamental point of the study is not properly addressed. In conclusion, high resolution by Instant Structured Illumination Microscopy presents lacks in results and methodologies for mesenchymal-amoeboid transition, that is a second key point of the manuscript; but this process was not evaluated properly and no additional assays were performed to confirm mesenchymal-amoeboid transition in GBM after CCG-1423 treatment. Such a flaws prevents any conclusion to be made.

  1. Response: We thank the reviewer for pointing out the difficulty to follow the flow of the text. We have now improved the figures so that they are high resolution, have carried out additional parametric or non-parametric testing including ANOVA analysis and have added some additional data that shows that targeting the two main migratory pathways adopted by cancer cells, mesenchymal and amoeboid, in combination does reduce the migratory ability of the glioma cells studied here (see in the main text). We did not present the iSIM data in great detail as we have already reported on this in our previous publication where the methods were thoroughly described (see Ketchen S, Rohwedder A, Knipp S, Esteves F, Struve N, Peckham M, et al. A novel workflow for three-dimensional analysis of tumour cell migration. Interface Focus. 2020 Apr 6;10(2):20190070.).

Specific comments:

  1. In page 5 line 16, check if Figure 1A referred to CCN1 staining is correct. Caption of Figure 1 is very long and complex. I would suggest reducing the description adding more information in the results and materials and methods section. CCN1 should be added in axis y of figures 1.

Response: Figure 1 has now been improved as suggested, see new figure 1 and figure legend.

  1. Figure 2 has several flaws. Title is indicated as “Effects of CCG-1423 on CCN1 secretion”. This is not appropriate and coherent with the experimental activities and data showed regarding migration and morphological changes. The authors should also explain how the displacement and migration analysis is performed. There are no representative images to confirm the effects of migration, displacement and morphological changes. Time points in A and B are not showed. “Concentration of CCN1” should be added in axis y of figures 2c and d. Variation on CCN1 levels in the supernatants over time in figures 2c and 2d should be tested with statistical analysis among time points. The description of these results is not clear.

Response: Figure 2 has now also been changed to reflect the comments made by the reviewer. Migration and displacement have been previously described and is also defined in line 166 – 169 to make this clear.

  1. In page 8 lines 19 the reference to Figure 2B is not correct.

Response: This has been now corrected, see text referring to Figure 2B, line 285, 286.

  1. It is necessary the nuclear counterstaining in Figures 3 and merged images with nuclei. Moreover, three cells per ROI are not representative to confirm the results.

Response: The results here are representative examples of the experiments which we carried out. We feel that the images are better to understand the way they are presented here to highlight the shift in protein localisation – we have added more explanation in the text with regards to figure 3 (line 320/321).  

  1. Nuclear and cytoplasmatic MKL1 is compared in controls versus treated. It should be appropriate to compare also MKL1 Nuclear versus MKL1 cytoplasmatic in controls and MKL1 Nuclear versus MKL1 cytoplasmatic in treated. In western blot analysis MKL1 accumulation in the cytoplasm should be also validated comparing samples among 24 hrs, 48 hrs and 72 hrs (i.e., MKL1 band intensity of CGC-1423 over time; the same for control group). This point regards also supplementary figure 1.

Response: We don’t fully understand what it is asked here. We have updated the text and hope that our workings here are now clearer.

  1. In page 9, lines 17-20. It is not clear the meaning, the interpretation and the reference about the results on live cell imaging.

Response: We have now rewritten the part relating to the results from the live cell imaging (line 276 -297).

  1. The authors wrote “The observed cytoplasmic build-up of MKL1 suggests the induction of MAT in glioma cells after treatment with CCG-1423”. According the previously results, the formulation of this hypothesis is not clear. Indeed, the authors showed that MKL1 accumulation on cytoplasm was associated to the CCG-1423 treatment. They also reported MKL1stimulates migration by CCN1 production after its nuclear translocation. Therefore, why should the accumulation of MLK1 provide an evidence of MAT induction?

Response: We have now reworded out hypothesis to make this clearer - see line 389 – 408.

  1. Figure 4 show a single cell. This is not representative to confirm the results. Moreover, color bar is not visible. Staining with Alexa Fluor 488 phalloidin is not reported in materials and methods.

Response: We decided to highlight the effect here we observed using one cell as we had already commented on this phenotype in a previous publication (see Ketchen et al as above response 1). We have added some additional data to show that combination treatment of U87 spheroids treated with CCG-1423 and Rhosin hydrochloride had anti-migratory activity on this cell line indicating that targeting the two main migratory pathways is important.

  1. The effects of inhibitors on cell migration and CCN1 secretion was evaluated in U251-MG and U87-MG cells lines. Why do the cytoplasmic MKL1 localization and MAT induction were evaluated only in U251-MG cell lines?

Response: We focussed on U251 as this cell line is the more representative cell line in terms of mesenchymal migration (velocity, generation of extensive lamellipodia and..); this has been included in the text (line 314 - 321). In addition, we saw comparable results for U87 and U251. This has also been added to the text (line 314 – 321).

Minor issues:

  1. English language is appropriate and understandable. I suggest doing minor check of English. Some sentences have an incorrect syntax. The conjunctions or punctuation are missing and the sentences are not linked correctly. Reading is not fluent in two examples below:

Page 1, Lines 40-43: “Nearly all patients will develop a recurrent tumour, despite therapy, and treatment options are limited to palliative approaches”.

Response: We have now corrected this, see line 43 – 44.

Page 2, Lines 7-9: “In addition, in vivo in immunodeficient mice implanted with, for example, human gastric adenocarcinoma cells, overexpression of CCN1 is an effective inducer of angiogenesis, concomitant with tumour vascularisation and growth”

We have now changed this sentence in the text (line 56 - 59).

  1. A long list of P values ​​is found within the main result text. I suggest removing them all since the statistical analysis is shown in figures and their presence in the text confuses the reading.

Response: We have improved all the figures and associated figure legends, please see in text.

Reviewer 3 Report

The submitted manuscript described a drug resistance mechanism in which GBM undergoes MAT upon treatment of CCG-1423. Authors illustrated how CCG-1423 inhibited MLK1 activities and related CCN1 expression. At last, authors claimed that continued migration after CCG-1423 treatment was due to MAT in glioblastoma. However, there are several points below need correction before the manuscript being considered for publication.

1) Scale bars should be presented in Figure 1E

2) How were error bars (technical repeats?) in Fig1D were obtained? Authors shall describe in the figure caption part.

2) Page 7 line 23, authors described cell rounding data in the text of part 3.3. However, the authors did not call any figure or data to back the claim.

3)Authors shall double check all their contexts in part 3.3. It appears that the figure callings are all wrong: In page 8 line 17, authors stated “The concentration of CCN1 in the medium did not increase in the treated cells, whereas there was a detectable increase in CCN1 levels in the supernatants from untreated cells over time (Figure 2B)”. However, Figure2B is migration data, which is totally irrelevant. Do authors mean 2C or 2D? Figure 2C and 2D were never called in the manuscript.

4) In discussion or conclusion section, I suggest authors point out weakness of the study:

i)CCG-1423 targets MKL-1 that mediates expression of many downstream targets. CCN1 is just one of them. The whole Rho signaling pathways were not investigated. Authors shall point out in future perspective that other MKL-1 targets should be investigated to further verify the mechanism.  

ii)The manuscript only reported an indirect method to show MAT using iSIM. Future direction shall include other in vitro methods to such as characterization of RhoA activation and MMP activity. (see  Chikina A.S., Alexandrova A.Y. (2018) An In Vitro System to Study the Mesenchymal-to-Amoeboid Transition. In: Gautreau A. (eds) Cell Migration. Methods in Molecular Biology, vol 1749. Humana Press, New York, NY.)

Author Response

The submitted manuscript described a drug resistance mechanism in which GBM undergoes MAT upon treatment of CCG-1423. Authors illustrated how CCG-1423 inhibited MLK1 activities and related CCN1 expression. At last, authors claimed that continued migration after CCG-1423 treatment was due to MAT in glioblastoma. However, there are several points below need correction before the manuscript being considered for publication.

We thank this reviewer for his comments and interest in our work.

  • Scale bars should be presented in Figure 1E

Response: We have added the magnification to the figure legend.

  • How were error bars (technical repeats?) in Fig1D were obtained? Authors shall describe in the figure caption part.

Response: The samples were added in duplicate as per manufacturer’s instructions.

  • Page 7 line 23, authors described cell rounding data in the text of part 3.3. However, the authors did not call any figure or data to back the claim.

Response: We have now rewritten the part relating to the cell rounding in the text (see lines 404-407).

  • Authors shall double check all their contexts in part 3.3. It appears that the figure callings are all wrong: In page 8 line 17, authors stated “The concentration of CCN1 in the medium did not increase in the treated cells, whereas there was a detectable increase in CCN1 levels in the supernatants from untreated cells over time (Figure 2B)”. However, Figure2B is migration data, which is totally irrelevant. Do authors mean 2C or 2D? Figure 2C and 2D were never called in the manuscript.

Response: We have now corrected the figure and the figure legend, see in manuscript.

4) In discussion or conclusion section, I suggest authors point out weakness of the study:

i)CCG-1423 targets MKL-1 that mediates expression of many downstream targets. CCN1 is just one of them. The whole Rho signaling pathways were not investigated. Authors shall point out in future perspective that other MKL-1 targets should be investigated to further verify the mechanism.  

Response: This is a very valid point; we have now included this in the discussion (line 445 – 450).

ii)The manuscript only reported an indirect method to show MAT using iSIM. Future direction shall include other in vitro methods to such as characterization of RhoA activation and MMP activity. (See Chikina A.S., Alexandrova A.Y. (2018) An In Vitro System to Study the Mesenchymal-to-Amoeboid Transition. In: Gautreau A. (eds) Cell Migration. Methods in Molecular Biology, vol 1749. Humana Press, New York, NY.)

Response: We thank the reviewer to point out this interesting literature, we have included this in the discussion (see line 445 – 450).

Round 2

Reviewer 1 Report

The authors addressed all my questions. The manuscript can be accepted in present form.

Author Response

We thank the reviewer for the final approval.

Reviewer 2 Report

Quality of presentation has now been improved and results are written more properly than the previous version of the manuscript. There are still some imprecisions and unsolved questions. 

  1. Figures 2A-B: There are no representative images to confirm the effects of migration and displacement.
  2. Figures 2C-D: are there statistical significative changes over time?
  3. Figures 3A-B: it is no clear the reason why the authors does not shown cells with the DAPI despite the nucleus stain has been indicated in the materials and methods. Moreover, the authors do not motivate properly the choice of showing just 3 cells as representative images for such an important experiment. The same for Figure 4 (only 1 cell).
  4. Figures 3D: are there statistical significative changes over time? The same question is for Suppl. Fig. 1.
  5. Why are the results of Figures 4 included in the discussion and not in a section of the results?
  6. Lines 401-405: It is not clear. Do the authors show results (about Figure 4A) that are referred to this dataset or these are obtained from a previous work? Do they indicate that the quantifications are included in the previous work and not in this manuscript? They wrote “a quantification of the data is shown in fig. 3 of the publication”. Which publication do they refer to? Please, clarify this point.
  7. Lines 439-443: The sentence is long and complex. Please, explain better the results.
  8. Lines 443-446: Is there a reference for these results?

Author Response

Response to the reviewer:

We have now addressed the reviewer’s comments, thank you for reviewing this work and feedback given:

  1. Figures 2A-B: There are no representative images to confirm the effects of migration and displacement.

Response: We have now added images with cell tracks from the original movies in supplemental figure 2.

  1. Figures 2C-D: are there statistical significative changes over time?

Response: There is now a reference to this in the figure legend.

3. Figures 3A-B: it is no clear the reason why the authors does not shown cells with the DAPI despite the nucleus stain has been indicated in the materials and methods. Moreover, the authors do not motivate properly the choice of showing just 3 cells as representative images for such an important experiment. The same for Figure 4 (only 1 cell).

Response: We have now included the DAPI stain in the figure. We chose to show another example of MAT we have discussed in previous work to highlight this phenomenon as part of this paper. We have responded to the very first set of comments by carrying out additional experiments within one month, which was very time consuming; therefore we have included here one representative example which we have marked as preliminary data. This is a very exciting development of this research which we are currently following up. We had quantified by technical repeats the effect of combination treatment on U87 in the work published by Hayley Butler, who carried out this work at the University of Huddersfield. This publication is cited in the text.

 4. Figures 3D: are there statistical significative changes over time? The same question is for Suppl. Fig. 1.

Response: We have now added this to the figure legend.

5. Why are the results of Figures 4 included in the discussion and not in a section of the results.

Response: We felt this was better for the flow of the paper but have now changed this. Please see highlighted text in the manuscript.

6. Lines 401-405: It is not clear. Do the authors show results (about Figure 4A) that are referred to this dataset or these are obtained from a previous work? Do they indicate that the quantifications are included in the previous work and not in this manuscript? They wrote “a quantification of the data is shown in fig. 3 of the publication”. Which publication do they refer to? Please, clarify this point.

Response: This point is now clarified as indicated by the highlighted text in the manuscript.

7. Lines 439-443: The sentence is long and complex. Please, explain better the results.

Response: This has now been changed, please see highlighted text 453 – 457.

8. Lines 443-446: Is there a reference for these results?

Response: We have now referenced that cancer cells can adopt collective cell migration. Highlighted in line 461 and also in the references.

Reviewer 3 Report

Authors have addressed my concerns.

Author Response

We thank the reviewer for final approval.

Round 3

Reviewer 2 Report

I suggest these minor corrections:

  1. Add cell line and scale bar of supplementary Figure 2A.
  2. Specify the number of cells/ROI analyzed for migration and displacement.
  3. Is there a specific reason why images 3a are not merged with DAPI and Green fluorescent staining?
  4. May the absence of significative statistical differences in figures 2c-d, 3d and supplementary1  have an important impact regarding the results?

Author Response

Reply to reviewer:

We are thankful to the reviewer to point out this oversight from us. We have now added scalebars and labelling for cell lines to supplement figure 2.

The number of tracked cells per condition is now indicated in the same legend.(point 1 and 2).

To point 3):

As we have already pointed out previously, the DAPI labelling merged with the image illustrating MKL1 IF would mask the effect of the inhibitor, and would make it especially hard to see the low MKL1 signal in the nucleus of untreated cells. Thus, a separate DAPI image is given (after suggestion of the reviewer), but an additional merged image is left out to save space, as it could not give any further information.

4) We are a bit confused about this comment. The addressed quantifications showed no significant differences in the originally re-submitted versions of our manuscript. We have interpreted and discussed these results accordingly and treated them rather as indications and trends. Thus, any potential impact on the results is already included in the manuscript.